# Robust training of recurrent neural networks to handle missing data for disease progression modeling

**Mostafa Mehdipour Ghazi**[1,2,3]**, Mads Nielsen**[1,2]**, Akshay Pai**[1,2]**, M. Jorge Cardoso**[3]**,**
**Marc Modat**[3]**, Sebastien Ourselin**[3]**, Lauge Sørensen**[1,2]
[1]Biomediq A/S, Copenhagen, DK
[2]Department of Computer Science, Univeristy of Copenhagen, DK
[3]Centre for Medical Image Computing, University College Londen, UK
mehdipour@biomediq.com

## Abstract

Disease progression modeling (DPM) using longitudinal data is a challenging task in machine learning for healthcare that can provide clinicians with better tools for diagnosis and monitoring of disease. Existing DPM algorithms neglect temporal dependencies among measurements and make parametric assumptions about biomarker trajectories. In addition, they do not model multiple biomarkers jointly and need to align subjects' trajectories. In this paper, recurrent neural networks (RNNs) are utilized to address these issues. However, in many cases, longitudinal cohorts contain incomplete data, which hinders the application of standard RNNs and requires a pre-processing step such as imputation of the missing values. We, therefore, propose a generalized training rule for the most widely used RNN architecture, long short-term memory (LSTM) networks, that can handle missing values in both target and predictor variables. This algorithm is applied for modeling the progression of Alzheimer's disease (AD) using magnetic resonance imaging (MRI) biomarkers. The results show that the proposed LSTM algorithm achieves a lower mean absolute error for prediction of measurements across all considered MRI biomarkers compared to using standard LSTM networks with data imputation or using a regression-based DPM method. Moreover, applying linear discriminant analysis to the biomarkers' values predicted by the proposed algorithm results in a larger area under the receiver operating characteristic curve (AUC) for clinical diagnosis of AD compared to the same alternatives, and the AUC is comparable to state-of-the-art AUC's from a recent cross-sectional medical image classification challenge. This paper shows that built-in handling of missing values in LSTM network training paves the way for application of RNNs in disease progression modeling.

## 1 Introduction

Alzheimer's disease (AD) is a chronic neurodegenerative disorder that begins with short-term memory loss and develops over time, causing issues in conversation, orientation, and control of bodily functions [1]. Early diagnosis of the disease is challenging and the diagnosis is usually made once cognitive impairment has already compromised daily living. Hence, developing robust, data-driven methods for disease progression modeling (DPM) utilizing longitudinal data is necessary to yield a complete perspective of the disease for better diagnosis, monitoring, and prognosis [2].

Existing DPM techniques attempt to describe biomarker measurements as a function of disease progression through continuous curve fitting. In the AD progression literature, a variety of regression-based methods have been applied to fit logistic or polynomial functions to the longitudinal dynamic

1st Conference on Medical Imaging with Deep Learning (MIDL 2018), Amsterdam, The Netherlands.

of each biomarker [3–8]. However, parametric assumptions on the biomarker trajectories limit the applicability of such methods; in addition, none of the existing approaches considers the temporal dependencies among measurements. Furthermore, the available methods mostly rely on independent biomarker modeling and require alignment of subjects' trajectories – either as a pre-processing step or as part of the algorithm.

Recurrent neural networks (RNNs) are sequence learning based methods that can offer continuous, non-parametric, joint modeling of longitudinal data while taking temporal dependencies amongst measurements into account [9]. However, since longitudinal cohort data often contain missing values due to, for instance, dropped out patients, unsuccessful measurements, and/or varied trial design, standard RNNs require pre-processing steps for data imputation which may result in suboptimal analyses and predictions [10]. Therefore, the lack of methods to inherently handle incomplete data in RNNs is evident [11].

Long short-term memory (LSTM) networks are widely used types of RNNs developed to effectively capture long-term temporal dependencies by dealing with the exploding and vanishing gradient problem during backpropagation through time [12–14]. They employ a memory cell with nonlinear reset units – so called constant error carousels (CECs), and learn to store history for either long or short time periods. Since their introduction, a variety of LSTM networks have been developed for different time-series applications [15]. The vanilla LSTM, among others, is the most commonly used architecture that utilizes three reset gates with full gate recurrence and applies backpropagation algorithm through time using full gradients. Nevertheless, its complete topology can include biases and cell-to-gates (peephole) connections.

The most common approach to handling missing data with LSTM networks is data interpolation pre-processing step, usually using mean or forward imputation. This two-step procedure decouples missing data handling and network training, resulting in a sub-optimal performance, and it is heavily influenced by the choice of data imputation scheme. Other approaches, update the architecture to utilize possible correlations between missing values' patterns and the target to improve prediction results [10, 11]. Our goal is different; we want to make the training of LSTM networks robust to missing values to more faithfully capture the true underlying signal, and to make the learned model generalizable across cohorts – not relying on specific cohort or demographic circumstances correlated with the target.

In this paper, we propose a generalized method for training LSTM networks that can handle missing values in both target and predictor variables. This is achieved via applying the batch gradient descent algorithm together with normalizing the loss function and its gradients with respect to the number of missing points in target and input, to ensure a proportional contribution of each weight per epoch. The proposed LSTM algorithm is applied for modeling the progression of AD in the Alzheimer's Disease Neuroimaging Initiative (ADNI) cohort [16] based on magnetic resonance imaging (MRI) biomarkers, and the estimated biomarker values are used to predict the clinical status of subjects.

Our main contribution is three-fold. Firstly, we propose a generalized formulation of backpropagation through time for LSTM networks to handle incomplete data and show that such built-in handling of missing values provides better modeling and prediction performances compared to using data imputation with standard LSTM networks. Secondly, we model temporal dependencies among measurements within the ADNI data using the proposed LSTM network via sequence-to-sequence learning. To the best of our knowledge, this is the first time such multi-dimensional sequence learning methods are applied for neurodegenerative DPM. Lastly, we introduce an end-to-end approach for modeling the longitudinal dynamics of imaging biomarkers – without need for trajectory alignment – and for clinical status prediction. This is a practical way to implement a robust DPM for both research and clinical applications.

## 2 Proposed LSTM algorithm

The main goal of this study is to minimize the influence of missing values on the learned LSTM network parameters. This is achieved by using the batch gradient descend scheme together with the backpropagation through time algorithm modified to take into account missing data in the input and target vectors. More specifically, the algorithm accumulates the input weight gradients proportionally weighted according to the number of available time points per input biomarker node using the subject-specific normalization factor of $\beta_n^j$. In addition, it uses an L2-norm loss function with residuals

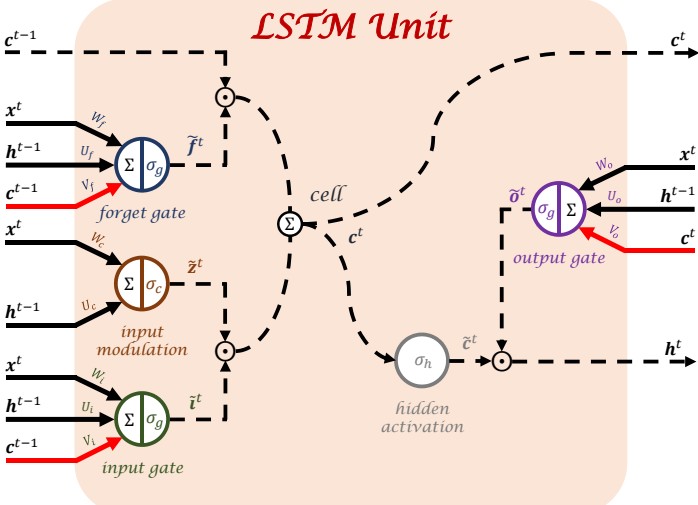

Figure 1: An illustration of a vanilla LSTM unit with peephole connections in red. The solid and dashed lines show weighted and unweighted connections, respectively.

weighted according to the number of available time points per output biomarker node using the subject-specific normalization factor of $\beta_m^j$, and normalized with respect to the total number of available input values for all visits of all biomarkers – propagated through the forward pass – using the subject-specific normalization factor of $\beta_x^j$. Such modification of the loss function also ensures that all gradients of the network weights are indirectly normalized. Finally, the use of batch gradient descend ensures that there is at least one visit available per biomarker so that each input node can proportionally contribute in the weight updates.

## 2.1 The basic LSTM architecture

Figure 1 shows a typical schematic of a vanilla LSTM architecture. As can be seen, the topology includes a memory cell, an input modulation gate, a hidden activation function, and three nonlinear reset gates, namely input gate, forget gate, and output gate, each of which accepting current and recurrent inputs. The memory cell learns to maintain its state over time while the multiplicative gates learn to open and close access to the constant error/information flow, to prevent exploding or vanishing gradients. The input gate protects the memory contents from perturbation by irrelevant inputs, while the output gate protects other units from perturbation by currently irrelevant memory contents. The forget gate deals with continual or very long input sequences, and finally, peephole connections allow the gates to access the CEC of the same cell state.

## 2.2 Feedforward in LSTM networks

Assume $\boldsymbol{x}_j^t \in \mathbb{R}^{N \times 1}$ is the $j$-th observation of an $N$-dimensional input vector at current time $t$. If $M$ is the number of output units, feedforward calculations of the LSTM network under study can be summarized as

$$\boldsymbol{f}_j^t = W_f \boldsymbol{x}_j^t + U_f \boldsymbol{h}_j^{t-1} + \boldsymbol{V}_f \odot \boldsymbol{c}_j^{t-1} + \boldsymbol{b}_f \longrightarrow \tilde{\boldsymbol{f}}_j^t = \sigma_g(\boldsymbol{f}_j^t),$$

$$\boldsymbol{i}_j^t = W_i \boldsymbol{x}_j^t + U_i \boldsymbol{h}_j^{t-1} + \boldsymbol{V}_i \odot \boldsymbol{c}_j^{t-1} + \boldsymbol{b}_i \longrightarrow \tilde{\boldsymbol{i}}_j^t = \sigma_g(\boldsymbol{i}_j^t),$$

$$\boldsymbol{z}_j^t = W_c \boldsymbol{x}_j^t + U_c \boldsymbol{h}_j^{t-1} + \boldsymbol{b}_c \longrightarrow \tilde{\boldsymbol{z}}_j^t = \sigma_c(\boldsymbol{z}_j^t),$$

$$\boldsymbol{c}_j^t = \tilde{\boldsymbol{f}}_j^t \odot \boldsymbol{c}_j^{t-1} + \tilde{\boldsymbol{i}}_j^t \odot \tilde{\boldsymbol{z}}_j^t \longrightarrow \tilde{\boldsymbol{c}}_j^t = \sigma_h(\boldsymbol{c}_j^t),$$

$$\boldsymbol{o}_j^t = W_o \boldsymbol{x}_j^t + U_o \boldsymbol{h}_j^{t-1} + \boldsymbol{V_o} \odot \boldsymbol{c}_j^t + \boldsymbol{b}_o \longrightarrow \tilde{\boldsymbol{o}}_j^t = \sigma_g(\boldsymbol{o}_j^t),$$

$$\boldsymbol{h}_j^t = \tilde{\boldsymbol{o}}_j^t \odot \tilde{\boldsymbol{c}}_j^t,$$

where $\{\boldsymbol{f}_j^t, \boldsymbol{i}_j^t, \boldsymbol{z}_j^t, \boldsymbol{c}_j^t, \boldsymbol{o}_j^t, \boldsymbol{h}_j^t\} \in \mathbb{R}^{M \times 1}$ and $\{\tilde{\boldsymbol{f}}_j^t, \tilde{\boldsymbol{i}}_j^t, \tilde{\boldsymbol{z}}_j^t, \tilde{\boldsymbol{c}}_j^t, \tilde{\boldsymbol{o}}_j^t\} \in \mathbb{R}^{M \times 1}$ are $j$-th observation of forget gate, input gate, modulation gate, cell state, output gate, and hidden output at time $t$ before and after activation, respectively. Moreover, $\{W_f, W_i, W_o, W_c\} \in \mathbb{R}^{M \times N}$ and $\{U_f, U_i, U_o, U_c\} \in \mathbb{R}^{M \times M}$ are sets of connecting weights from input and recurrent, respectively, to the gates and cell, $\{\boldsymbol{V}_f, \boldsymbol{V}_i, \boldsymbol{V}_o\} \in \mathbb{R}^{M \times 1}$ is the set of peephole connections from the cell to the gates, $\{\boldsymbol{b}_f, \boldsymbol{b}_i, \boldsymbol{b}_o, \boldsymbol{b}_c\} \in \mathbb{R}^{M \times 1}$ represents corresponding biases of neurons, and $\odot$ denotes element-wise multiplication. Finally, $\sigma_g$, $\sigma_c$, and $\sigma_h$ are nonlinear activation functions assigned for the gates, input modulation, and hidden output, respectively. Logistic sigmoid functions are applied for the gates with range $[0, 1]$ while hyperbolic tangent functions are applied for modulation of both cell input and hidden output with range $[-1, 1]$.

## 2.3 Robust backpropagation through time

Let $\mathcal{L} \in \mathbb{R}^{M \times 1}$ be the loss function defined based on the actual target $\boldsymbol{s}$ and network output $\boldsymbol{y}$. Here, we consider one layer of LSTM units for sequence learning which means that the network output is the hidden output. The main idea is to calculate the partial derivatives of the normalized loss function ($\delta$) with respect to the weights using the chain rule. Hence, the backpropagation calculations through time using full gradients can be obtained as

$$\mathcal{L}(m) = \frac{1}{2}\sum_{j,t}\frac{1}{\beta_x^j\beta_m^j}(\boldsymbol{y}_j^t(m) - \boldsymbol{s}_j^t(m))^2 \longrightarrow \delta\boldsymbol{y}_j^t(m) = \frac{\partial\mathcal{L}_j^t(m)}{\partial\boldsymbol{y}_j^t(m)} = \frac{1}{\beta_x^j\beta_m^j}(\boldsymbol{y}_j^t(m) - \boldsymbol{s}_j^t(m)),$$

$$\delta\boldsymbol{h}_j^t = \delta\boldsymbol{y}_j^t + U_f^T\delta\boldsymbol{f}_j^{t+1} + U_i^T\delta\boldsymbol{i}_j^{t+1} + U_c^T\delta\boldsymbol{z}_j^{t+1} + U_o^T\delta\boldsymbol{o}_j^{t+1},$$

$$\delta\tilde{\boldsymbol{o}}_j^t = \delta\boldsymbol{h}_j^t \odot \tilde{\boldsymbol{c}}_j^t \longrightarrow \delta\boldsymbol{o}_j^t = \delta\tilde{\boldsymbol{o}}_j^t \odot \sigma_g'(\boldsymbol{o}_j^t),$$

$$\delta\tilde{\boldsymbol{c}}_j^t = \delta\boldsymbol{h}_j^t \odot \tilde{\boldsymbol{o}}_j^t \longrightarrow \delta\boldsymbol{c}_j^t = \delta\tilde{\boldsymbol{c}}_j^t \odot \sigma_h'(\boldsymbol{c}_j^t) + \delta\boldsymbol{c}_j^{t+1} \odot \tilde{\boldsymbol{f}}_j^{t+1} + \boldsymbol{V}_f \odot \delta\boldsymbol{f}_j^{t+1} + \boldsymbol{V}_i \odot \delta\boldsymbol{i}_j^{t+1} + \boldsymbol{V}_o \odot \delta\boldsymbol{o}_j^t,$$

$$\delta\tilde{\boldsymbol{z}}_j^t = \delta\boldsymbol{c}_j^t \odot \tilde{\boldsymbol{i}}_j^t \longrightarrow \delta\boldsymbol{z}_j^t = \delta\tilde{\boldsymbol{z}}_j^t \odot \sigma_c'(\boldsymbol{z}_j^t),$$

$$\delta\tilde{\boldsymbol{i}}_j^t = \delta\boldsymbol{c}_j^t \odot \tilde{\boldsymbol{z}}_j^t \longrightarrow \delta\boldsymbol{i}_j^t = \delta\tilde{\boldsymbol{i}}_j^t \odot \sigma_g'(\boldsymbol{i}_j^t),$$

$$\delta\tilde{\boldsymbol{f}}_j^t = \delta\boldsymbol{c}_j^t \odot \boldsymbol{c}_j^{t-1} \longrightarrow \delta\boldsymbol{f}_j^t = \delta\tilde{\boldsymbol{f}}_j^t \odot \sigma_g'(\boldsymbol{f}_j^t),$$

$$\delta\boldsymbol{x}_j^t = W_f^T\delta\boldsymbol{f}_j^t + W_i^T\delta\boldsymbol{i}_j^t + W_c^T\delta\boldsymbol{z}_j^t + W_o^T\delta\boldsymbol{o}_j^t,$$

where $\beta_x^j = J\frac{|\boldsymbol{x}_j|}{TN}$ and $\beta_m^j = |\boldsymbol{y}_j(m)|$ are normalization factors to handle missing values of the $j$-th observation with batch size $J$ and sequence length $T$. Also, $|\boldsymbol{x}_j|$ and $|\boldsymbol{y}_j(m)|$ denote the total number of available input values and the number of available target time points in the $m$-th node, respectively. Finally, if $\theta \in \{f, i, z, o\}$ and $\phi \in \{f, i\}$, the gradients of the loss function with respect to the weights are calculated as

$$\delta W_\theta(n) = \sum_{j=1}^{J}\frac{1}{\beta_n^j}\delta\boldsymbol{\theta}_j^{\{0 \to T\}}\boldsymbol{x}_j^{\{0 \to T\}}(n),$$

$$\delta U_\theta = \sum_{j=1}^{J}\delta\boldsymbol{\theta}_j^{\{1 \to T\}}\boldsymbol{h}_j^{\{0 \to T-1\}},$$

$$\delta\boldsymbol{V}_\phi = \sum_{j=1}^{J}\sum_{t=0}^{T-1}\delta\phi_j^{t+1} \odot \boldsymbol{c}_j^t,$$

$$\delta\boldsymbol{V}_o = \sum_{j=1}^{J}\sum_{t=0}^{T}\delta\boldsymbol{o}_j^t \odot \boldsymbol{c}_j^t,$$

$$\delta\boldsymbol{b}_\theta = \sum_{j=1}^{J}\sum_{t=0}^{T}\delta\boldsymbol{\theta}_j^t,$$

Table 1: Demographics statistics of the TADPOLE dataset

| | Number of visits | | Age, year (mean±SD) | | Education, year |
| | male | female | male | female | (mean±SD) |
| --- | --- | --- | --- | --- | --- |
| CN | 1,356 | 1,389 | 76.67±6.44 | 75.85±6.28 | 16.38±2.70 |
| MCI | 2,454 | 1,604 | 75.59±7.47 | 73.87±8.09 | 15.91±2.84 |
| AD | 1,208 | 900 | 77.22±7.11 | 75.45±7.92 | 15.18±2.99 |
| All (labeled & unlabeled) | 12,741 | | 76.00±7.38 | | 15.91±2.86 |

where $\beta_n^j = \frac{|\boldsymbol{x}_j(n)|}{T}$ is the normalization factor handling missing input values and $|\boldsymbol{x}_j(n)|$ is the number of available input time points in the $n$-th node.

### 2.4 Momentum batch gradient descent

As an efficient iterative algorithm, momentum batch gradient descent is applied to find the local minimum of the loss function calculated over a batch while speeding up the convergence. The update rule can be written as

$$\vartheta^{new} = \mu\vartheta^{old} - \alpha(\delta\omega + \gamma\omega^{old}),$$
$$\omega^{new} = \omega^{old} + \vartheta^{new},$$

where $\vartheta$ is the weight update initialized to zero, $\omega$ is the to-be-updated weight array, $\delta\omega$ is the gradient of the loss function with respect to $\omega$, and $\alpha$, $\gamma$, and $\mu$ are the learning rate, weight decay or regularization factor, and momentum weight, respectively.

## 3 Experiments

### 3.1 Data preparation

We utilize the dataset from The Alzheimer's Disease Prediction Of Longitudinal Evolution [1] [17] (TADPOLE) challenge for DPM using the LSTM network. The dataset is composed of data from the three ADNI phases ADNI 1, ADNI GO, and ADNI 2. This includes roughly 1,500 biomarkers acquired from 1,737 subjects (957 males and 780 females) during 12,741 visits at 22 distinct time points between 2003 and 2017. Table 1 summarizes statistics of the demographics in the TADPOLE dataset. Note that the subjects include missing measurements during their visits and not all of them are clinically labeled.

In this work, we have merged existing groups labeled as cognitively normal (CN), significant memory concern (SMC), and normal (NL) under CN, mild cognitive impairment (MCI), early MCI (EMCI), and late MCI (LMCI) under MCI, and Alzheimer's disease (AD) and Dementia under AD. Moreover, groups with labels converting from one status to another, e.g. "MCI-to-AD", are assumed to belong the next status ("AD" in this example).

MRI biomarkers are used for AD progression modeling. This includes T1–weighted brain MRI volumes of ventricles, hippocampus, whole brain, fusiform, middle temporal gyrus, and entorhinal cortex. We normalize the MRI measurements with respect to the corresponding intracranial volume (ICV). Out of 22 visits, we select 11 visits – including baseline – with a fix interval of one year to span the majority of measurements and subjects. Next, we filter data outliers based on the specified range of each biomarker and normalize the measurements to be in the range $[-1, 1]$. Finally, subjects with less than three distinct visits for any biomarker are removed to obtain 742 subjects. This is to ensure that at least two visits are available per biomarker for performing sequence learning through the feedforward step and an additional visit for backpropagation.

For evaluation purpose, we partition the entire dataset to three non-overlapping subsets for training, validation, and testing. To achieve this, we randomly select 10% of the within-class subjects for

---

[1]https://tadpole.grand-challenge.org

validation and the same for testing. More specifically, based on the baseline labels of subjects, we randomly pick within-class samples ensuring to have enough subjects with few and large number of visits in each subset. This process results in 592, 76, and 74 subjects for training, validations, and testing, respectively.

## 3.2  Evaluation metrics

Mean absolute error (MAE) and multi-class area under the receiver operating characteristic (ROC) curve (AUC) are used to assess the modeling and classification performances, respectively. MAE measures accuracy of continuous prediction per biomarker by computing the difference between actual and estimated values as follows

$$\text{MAE} = \frac{1}{\mathcal{I}} \sum_{j,t} |\boldsymbol{y}_j^t - \boldsymbol{s}_j^t|,$$

where $\boldsymbol{s}_j^t$ and $\boldsymbol{y}_j^t$ are the ground-truth and estimated values of the specific biomarker for the $j$-th subject at the $t$-th visit, respectively, and $\mathcal{I}$ is the number of existing points in the target array $\boldsymbol{s}$. Multi-class AUC [18], on the other hand, is a measure to examine the diagnostic performance in a multi-class test set using ROC analysis. It can be calculated using the posterior probabilities as follows

$$\text{AUC} = \frac{1}{(n_c(n_c-1))} \sum_{i=1}^{n_c-1} \sum_{k=i+1}^{n_c} \frac{1}{n_i n_k} \Big[ \text{SR}_i - \frac{n_i(n_i+1)}{2} + \text{SR}_k - \frac{n_k(n_k+1)}{2} \Big],$$

where $n_c$ is the number of distinct classes, $n_i$ denotes the number of available points belonging to the $i$-th class, and $\text{SR}_i$ is the sum of the ranks of posteriors $p(c_i|\boldsymbol{s}_i)$ after sorting all concatenated posteriors $\{p(c_i|\boldsymbol{s}_i), p(c_i|\boldsymbol{s}_k)\}$ in an increasing order, where $\boldsymbol{s}_i$ and $\boldsymbol{s}_k$ are vectors of scores belonging to the true classes $c_i$ and $c_k$, respectively.

## 3.3  Experimental setup

All the evaluated methods in this study are developed in-house in MATLAB R2017b and run on a 2.80 GHz CPU with 16 GB RAM. We initialize the LSTM network weights by generating uniformly distributed random values in the range $[-0.05, 0.05]$ and set the weights' updates and weights' gradients to zero. We set the batch size to the number of available training subjects. Furthermore, for simplicity, we use the first ten visits to estimate the second to eleventh visits per subject and use the estimated values for evaluation. Finally, we train the network using feedforward and the proposed method of backpropagation through time where the network replace the input missing values and corresponding error of the output missing values with zero.

We utilize the validation set to tune the network optimization parameters each time by adjusting one of the parameters while keeping the rest at fixed values to achieve the lowest average MAE. Peephole connections are used in the network as they intend to improve the performance. Based on these strategies, the optimal parameters are obtained as $\alpha = 0.1$, $\mu = 0.9$, and $\gamma = 0.0001$ with 1,000 epochs. The corresponding MAE's for the validation set are also calculated as $2.9590 \times 10^{-3}$, $2.4603 \times 10^{-4}$, $1.4943 \times 10^{-2}$, $2.4161 \times 10^{-4}$, $7.5522 \times 10^{-4}$, $9.6592 \times 10^{-4}$, respectively for ventricles, hippocampus, whole brain, entorhinal cortex, fusiform, and middle temporal gyrus. Moreover, it takes about 340 seconds and 0.025 seconds for training and validation, respectively. It is worthwhile mentioning that all the estimated biomarker's measurements are transformed back to their actual ranges while calculating MAE's.

## 3.4  Results

After successfully training our LSTM network, we examine it using the obtained test subset. Next, we train the network using mean imputation (LSTM-Mean) [11] and forward imputation (LSTM-Forward) [10]. Moreover, we use the parametric, regression-based method of [3] to model the AD progression. Table 2 compares the test modeling performance (MAE) of the MRI biomarkers using aforementioned approaches. As it can be deduced from Table 2, our proposed method outperforms all

Table 2: Test modeling performance (MAE) of the MRI biomarkers using different DPM methods.

| | Proposed | LSTM-Mean [11] | LSTM-Forward [10] | Jedynak et al. [3] |
|---|---|---|---|---|
| Ventricles | $3.0674 \times 10^{-3}$ | $6.2010 \times 10^{-3}$ | $4.7204 \times 10^{-3}$ | $8.0718 \times 10^{-3}$ |
| Hippocampus | $2.3267 \times 10^{-4}$ | $5.0916 \times 10^{-4}$ | $3.3977 \times 10^{-4}$ | $5.1455 \times 10^{-4}$ |
| Whole brain | $1.3298 \times 10^{-2}$ | $2.3746 \times 10^{-2}$ | $1.6389 \times 10^{-2}$ | $5.5125 \times 10^{-3}$ |
| Entorhinal cortex | $2.1138 \times 10^{-4}$ | $3.0324 \times 10^{-4}$ | $2.5489 \times 10^{-4}$ | $3.4660 \times 10^{-4}$ |
| Fusiform | $6.7932 \times 10^{-4}$ | $1.2964 \times 10^{-3}$ | $1.0044 \times 10^{-3}$ | $9.0342 \times 10^{-4}$ |
| Middle temporal gyrus | $8.6750 \times 10^{-4}$ | $1.2606 \times 10^{-3}$ | $1.1759 \times 10^{-3}$ | $1.1092 \times 10^{-3}$ |

Table 3: Test diagnostic performance (AUC) of the MRI biomarkers using LDA with different DPM methods.

| | Proposed | LSTM-Mean [11] | LSTM-Forward [10] | Jedynak et al. [3] |
|---|---|---|---|---|
| CN vs. MCI | 0.5914 | 0.5838 | 0.5800 | 0.5468 |
| CN vs. AD | 0.9029 | 0.8404 | 0.8150 | 0.7826 |
| MCI vs. AD | 0.7844 | 0.6936 | 0.6890 | 0.7330 |
| CN vs. MCI vs. AD | 0.7596 | 0.7059 | 0.6947 | 0.6875 |

other modeling techniques in all categories. It should be noticed that when we apply data imputation, the backpropagation formulas simply generalize to the standard LSTM network.

To assess the ability of the estimated biomarkers' measurements in predicting the clinical labels, we apply a linear discriminant analysis (LDA) classifier to the multi-dimensional training data estimations to compute the posterior probability scores in the test data. The obtained scores are then used to calculate the AUC's. The diagnostic prediction results for the test set are shown in Table 3 for the utilized methods. As can be seen, the proposed method outperforms all other schemes in predicting clinical status of subjects per visits. This, in turn, reveals the effect of modeling on classification performance. One could of course use different classifiers to improve the results. But our focus in this paper is on DPM or sequence-to-sequence learning. On the other hand, it is possible to train the LSTM network for a classification (sequence-to-label) problem. However, since this approach requires labeled data, it would only be able to use a subset of the utilized data in training.

Furthermore, the diagnostic classification results of the predicted MRI biomarkers' measurements using the proposed approach are comparable to state-of-the-art cross-sectional MRI-based classification results in the recent challenge on Computer-Aided Diagnosis of Dementia (CADDementia) [19]. To be more specific, LDA classification on predicted features using the proposed method achieves a multi-class AUC of 0.76 which is within the top-five multi-class AUCs in the challenge that ranged from 0.79 to 0.75.

## 4   Summary and discussion

In this paper, a training algorithm was proposed for LSTM networks aiming to improve robustness against missing data, and the robustly trained LSTM network was applied for AD progression modeling using longitudinal measurements of imaging biomarkers. To the best of our knowledge this is the first time RNNs have been studied and applied for DPM within the ADNI cohort. The proposed training method demonstrated better performance than using imputation prior to a standard LSTM network and outperformed an established parametric, regression-based DPM method, in terms of both biomarker prediction and subsequent diagnostic classification.

Moreover, the classification results using the predicted MRI measurements of the proposed method are comparable to those of the CADDementia challenge. It should, however, be noted that there are important differences between this study and the CADDementia challenge. Firstly, this work has the advantage of training and testing features from the same cohort whereas CADDementia algorithms were applied to classify data from independent cohorts. Secondly, the top performing CADDementia algorithms incorporated different types of MRI features besides volumetry. Thirdly,

in contrast to CADDementia where features were completely available, this work predicts features based on longitudinal data before classification.

This study highlights the potential of RNNs for modeling the progression of AD using longitudinal measurements, provided that proper care is taken to handle missing values and time intervals. In general, standard LSTM networks are designed to handle sequences with a fixed temporal or spatial sampling rate within longitudinal data. We used the same approach in the AD progression modeling application by disregarding, for example, visiting months 3, 6 and 18, and confining the experiments to yearly follow-up in the ADNI data. However, one could utilize modified LSTM architectures such as time-aware LSTM [20] to address irregular time steps in longitudinal patient records.

### Acknowledgments

This project has received funding from the European Union's Horizon 2020 research and innovation programme under the Marie Skłodowska-Curie grant agreement No 721820. This work uses the TADPOLE data sets https://tadpole.grand-challenge.org constructed by the EuroPOND consortium http://europond.eu funded by the European Union's Horizon 2020 research and innovation programme under grant agreement No 666992.

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
