# OpenReview forum: "Robust training of recurrent neural networks to handle missing data for disease progression modeling"
_MIDL.amsterdam/2018/Conference — MIDL 2018 Oral_

### Review · AnonReviewer3 · 2018-05-08
**A nice manuscript needing further refinement**

**Rating:** 2
**Confidence:** 2

**Review:**

This paper presents an RNN which appears more agnostic to missing data than standard practice.
While the paper is easy to follow, there are a number of points which weren’t addressed reducing the potential impact of the manuscript:

“Finally, subjects with less than three distinct visits per biomarker are removed to better adapt with the LSTM network for sequence learning.” -- > since this paper is essentially entirely about missing values and how they’re managed, the missing value component needs to be described fully:  how many patients had less than 3 visit, how many outliers were detected and removed (which I assume were then listed as missing values). What was the level of sparsity in the data? There is a huge difference between < 1% of the values missing and >50% of the values missing, but this manuscript doesn’t quantify this sufficiently.

The authors should empirically define the boundaries for the missing data imputation (e.g. MAE on the evaluation dataset for patients with < 10% missing data 10 - 20% missing data, etc). This would allow the reader to determine when traditional methods give an equivalent performance and when the RNN approach is merited (an interesting discussion for the discussion section).

There is no indication of the dimensionality of the dataset, how many biomarkers were captured for each patient? A good was of presenting this information would be to list all of the biomarkers in the dataset in a table, and then show how many of them were missing for the patients as a percent(e.g., 50% of patients had this particular biomarker missing at least once), in the case of multiple missing break it into 1 time missing, 3 or fewer times missing, 3 or more times missing.
Additionally describe the features in the dataset in a table (e.g. feature name, biological interpretation). If these data are published elsewhere, they should cite the original paper in the table (e.g. "adapted from [citation]")

Was the improvement in results statistically significant? Confidence intervals? Do increases in performance coincide with patients missing more or less data?

Summary and discussion: this section is normally reserved to summaries and discussion. Its inappropriate to add in what is essentially a new experimental comparison (CADDementia) in this location. That component should go in its own section dedicated to results. The 3rd paragraph seems like it belongs in the introduction section.

As opposed to solely reporting results at the cohort summary level, it would have been more informative if the authors pulled out singular patients, in particular ones which had the highest error and attempted to tease out why their approach performed poorly in those cases. Was there something special about them that their approach could not generalize to? Did other approaches perform well on those patients, motivating an ensemble approach? was there a particular trend in failures worth discussing, e.g., if the same biomarker was missing multiple times that performed worse than different biomarkers being absent a single time.

Having done such an analysis would have made the summary and discussion section easier to write as then they can discuss the algorithms performance, way of improvement, and insights that the authors have gained through their experiment which are of paramount importance to share with the lay reader.

Lastly, and of lesser importance for a conference paper like this but still needs some discussion, is to validate using a second dataset (of any variety) to show the robustness of the approach. It is not sufficient to show an improvement on a dataset score as a common issue with using challenge datasets is that the algorithms start to over fit to the particular dataset and don’t perform as well when using new data. Inherently, all readers want to know “will this work on my data set?”. For a journal version of this manuscript, such a concern would need to be addressed.


Is the code publically available? Can others test out this approach on their own data?



**Special Issue:**

Yes

---

### Review · AnonReviewer1 · 2018-05-09
**The paper is well written and clearly organized. Since the proposed method is general and not problem depended, it can be expected to make significant scientific and clinical impact.**

**Rating:** 5
**Confidence:** 2

**Review:**

#Summary of paper
This paper proposed a training algorithm for long term short memory (LSTM) networks aiming to improve robustness against missing time-sequence data. They demonstrated the application of their proposed method for the disease progress modeling of Alzheimer’s disease.

#Strength
-The proposed training algorithm is a general method that is not specified to particular problem in processing of time-sequence data.
- The paper introduces method for normalization of the loss function. This is a simple modification and seems to work well.
-In experiments, authors evaluated the performance of modeling and classification on a public challenge dataset. That is persuasive.

#Major weakness
-How many labels are missing in training dataset? For evaluation of the robustness against the missing data, a measurement of the level of “missing” is necessary.
#Minor weakness
-At each end of equations in successive, comma should be placed (pages. 3-5).
-Describing the processing time of learning and test is promising.


**Special Issue:**

Yes

---

### Review · AnonReviewer2 · 2018-05-09
**Modification of the training procedure for LSTM networks that can handle missing data**

**Rating:** 3
**Confidence:** 1

**Review:**

The authors present a modification of the training procedure for LSTM networks that can handle missing data and apply the proposed methodology for modelling the progression of Alzheimer’s disease.

I have limited experience in this application field, however, from I can tell the proposed methodological choices are reasonable well motivated.

The authors evaluate the proposed methodology on a public dataset using a good experimental setup and compare to three additional methods, two of which use data imputation to deal with the missing values problem.  Their algorithm has consistently better performance compared to the other three methods.

**Special Issue:**

No

---

### Comment · ~Bram_van_Ginneken1 · 2018-05-18
**Selection for longlist for special issue Medical Image Analysis**

Dear authors,

Congratulations on your acceptance to MIDL! We have selected your paper on the longlist for the Medical Image Analysis Special Issue. Please read this page:
https://midl.amsterdam/special-issue-in-medical-image-analysis/
Please answer the three questions that are listed on that page about your interest in submitting to the special issue, potential overlap with other publications, and related publications.

You can post your answer here directly below on openreview.net, or mail me directly at bram.vanginneken@radboudumc.nl.

Best regards, Bram

---

### Decision · Program_Chairs · 2018-05-15
**Paper98 Acceptance Decision**

Oral